# Influence of Age and Sex on the Pharmacokinetics of Midazolam and the Depth of Sedation in Pediatric Patients Undergoing Minor Surgeries

**DOI:** 10.3390/pharmaceutics15020440

**Published:** 2023-01-29

**Authors:** Carmen Flores-Pérez, Janett Flores-Pérez, Luis Alfonso Moreno-Rocha, Juan Luis Chávez-Pacheco, Norma Angélica Noguez-Méndez, Blanca Ramírez-Mendiola, Yolopsi Sánchez-Maza, Lina Sarmiento-Argüello

**Affiliations:** 1Pharmacology Laboratory, National Institute of Pediatrics, Mexico City 04530, Mexico; 2Doctorate in Biological and Health Sciences, Division of Biological and Health Sciences, Universidad Autónoma Metropolitana, Mexico City 04960, Mexico; 3Pharmacokinetics and Pharmacodynamics Laboratory, Division of Biological and Health Sciences, Universidad Autónoma Metropolitana, Mexico City 04960, Mexico; 4Molecular and Controlled Release Pharmacy Laboratory, Division of Biological and Health Sciences, Universidad Autónoma Metropolitana, Mexico City 04960, Mexico; 5Anesthesiology Department, INP, Mexico City 04530, Mexico

**Keywords:** midazolam, sedation, age, sex, pediatrics, minor surgeries

## Abstract

Whether age and sex influence the depth of sedation and the pharmacokinetics of midazolam is currently unknown. The influence of age and sex was investigated in 117 children (2 to 17 years) who required intravenous sedation for minor surgery (0.05 mg/kg). Plasma concentrations and sedation effects were simultaneously measured. The measured concentrations were analyzed using a two-compartment model with first-order elimination. Among the age ranges, significant differences were found (*p* < 0.05) between the volume of distribution (Vd) of the first compartment (V1) and that of the second (V2). With respect to sex, differences in V2 were found between age groups. At the administered dose, in patients younger than 6 years, a profound sedative effect (40–60 BIS) was observed for up to 120 min, while in older children, the effect lasted only half as long. The differences found in the Vd and bispectral index (BIS) in patients younger than 6 years compared to older patients may be due to immature CYP3A activity and body fat content; furthermore, the Vd varies with age due to changes in body composition and protein binding. Patients younger than 6 years require intravenous (IV) doses <0.05 mg/kg of midazolam for deep sedation. Dosage adjustments according to age group are suggested.

## 1. Introduction

Midazolam is used for premedication before the induction of anesthesia and to achieve conscious sedation during diagnostic or therapeutic procedures [1]. The IV dose of midazolam to induce sedation in children ranges from 0.025 to 0.1 mg/kg [2].

Midazolam has a rapid onset of action after IV administration because, at physiological pH, it is highly lipophilic and quickly crosses the blood–brain barrier, easily accessing the benzodiazepine receptors of the central nervous system [3]. It is metabolized by cytochrome P450 (CYP450) into several metabolites, including an active metabolite, alpha-hydroxymidazolam (1-OHMDZ) [4]. It is eliminated from the plasma almost exclusively by hepatic oxidative processes involving CYP450 [5] catalyzed by the CYP3A subfamily, which is involved in the oxidation of numerous drugs [6]. 1-OHMDZ represents 50–70% of midazolam metabolism [7], and the elimination half-life of the drug is generally 1.5–3.5 h [8].

The sedative potency of MDZ is approximately 3–4 times higher compared to diazepam. Thus, it has been associated with a higher level of amnesia and greater acceptability in adult patients with respect to diazepam [9,10,11,12]. MDZ allows greater sedation control and rapid recovery compared to other benzodiazepines, including diazepam, in children [13].

Respiratory depression, apnea, areflexia, respiratory and/or cardiac arrest (usually in combination with central nervous system depressant drugs), and episodes of hypotension may occur with overdosage [14]. Complications of insufficient sedation that have been reported are fear, anxiety and agitation, risk of recalling unpleasant situations or being aware of them, and the possibility of unintentional removal of medical devices [15].

Due to the observed differences in CYP3A4 expression and activity in the liver and intestine in different age groups, the clearance (Cl) of midazolam has been observed to be lower in children than in adults [11]. In children, the time to achieve a clinical effect is longer for midazolam than for any other sedative agent [8,13,16].

Differences in pharmacokinetics, particularly in the clearance of a drug, are known to occur due to factors such as age, weight, disease, and ethnicity or genotype [17]. Some studies also suggest sex differences in CYP3A activity, but results are inconsistent [18]. Age and sex differences in pharmacokinetics and pharmacodynamics include those related to physiology, such as body fat content and hormonal influences [19]; however, whether such differences influence the sedation of pediatric patients is unknown. Therefore, the objective of the present study was to identify differences related to age and sex in the pharmacokinetics of midazolam and the degree of sedation depth in pediatric patients undergoing minor surgery.

## 2. Materials and Methods

### 2.1. Classification of the Study

The study is descriptive, observational, prospective, and longitudinal. The present study is in accordance with the Declaration of Helsinki guidelines and approved by the Research, Biosafety, and Ethics Committee (IRB 00008064) of the National Institute of Pediatrics (INP-012/2019).

### 2.2. Patients

A total of 117 patients of both sexes aged 2–17 years were included and classified according to the American Society of Anesthesiologists (ASA) as ASA grades I and II. The patients received a single IV dose of 0.05 mg/kg of midazolam and underwent minor surgery at the INP from April 2019 to May 2022; a family member or responsible guardian consented to participate in the study by signing an informed consent form. Patients older than 7 years who agreed to participate in the study signed an informed assent form.

### 2.3. Perioperative Anaesthetic Management

The patients in this study included both hospitalized patients with peripheral venous access and outpatients who were referred to the operating room, and all patient documentation (including informed consent and/or assent) was confirmed to be compliant. Both groups were premedicated with midazolam at 0.05 mg/kg intravenously. All patients underwent basic monitoring: five-lead electrocardiogram, noninvasive blood pressure (children 2–5 years = 71.5 ± 8.8 mm Hg; 6–12 years = 77.5 ± 10.6 mm Hg; 13–17 years = 83.2 ± 11.3 mm Hg), pulse oximeter (individuals 2–5 years = 97.1 ± 1.8%; 6–12 years = 96.6 ± 2.4%; 13–17 years = 97.2 ± 1.8%), and a sensor to measure the bispectral index (BIS) and record its baseline value (patients 2–5 years = 90.6 ± 14.5; 6–12 years = 94.9 ± 5.3; 13–17 years = 94.2 ± 5.1). IV induction was performed for patients with peripheral venous access. Outpatients and those without peripheral venous access underwent inhalational induction with sevoflurane through a face mask. During inhalational induction, a peripheral venous route was established to administer 0.05 mg/kg midazolam, and then IV induction was initiated, allowing for subsequent anesthetic management with balanced general anesthesia and/or combined anesthesia.

Blood sampling was performed randomly three times per patient, considering the following times: 5, 10, 15, 20, 25, 30, 60, 90, and 120 min and the BIS value of each sample was documented. The samples obtained were coded and transferred to the Pharmacology Laboratory for subsequent analysis of midazolam concentrations by high-performance liquid chromatography (HPLC).

In the immediate postoperative period, the patients were monitored to ensure that they remained neurologically intact and hemodynamically stable with ventilatory automatism and without pain, nausea, or vomiting; tolerated oral intake; and left the postoperative care unit under the care of the treating service either in the hospital or on an outpatient basis.

To record possible adverse effects that may occur during treatment with midazolam, a format was established according to the guidelines of the Official Mexican Standard NOM-220-SSA1-2016 on the implementation and operation of pharmacovigilance [20].

### 2.4. Samples

Blood samples were collected in 3-mL Vacutainer™ tubes with heparin (Becton Dickinson, NJ, USA). From each tube, six aliquots were taken (each 50 μL), placed in circles containing Guthrie cards (903™, Marlab, Mexico), and left to dry at room temperature with protection from light for 6 h [21]. Subsequently, the cards were frozen at −80 °C (MDF-U76VC, Panasonic, Japan) until analysis.

### 2.5. Determination of Midazolam Concentrations in Blood

The samples were analyzed based on an HPLC method previously developed and validated in the Pharmacology Laboratory [22], with modification of the extraction procedure using dried blood drops or dried blood spots (DBS) in place of plasma. Two circles were cut from each Guthrie card (containing 50 µL of blood each) and placed in a test tube in triplicate, and the procedure was continued as described for the previously reported technique.

The analytical method was validated according to FDA and EMA international guidelines [23,24], testing for selectivity, linearity, reproducibility, repeatability, quantification limit, and stability to different conditions.

Midazolam hydrochloride, acetonitrile, and diethyl ether were purchased from Sigma (St Louis, MO, USA), potassium dihydrogen phosphate and sodium hydroxide were purchased from Merck (Darmstadt, Germany), and propranolol was purchased from ICN Biomedicals (OH, USA). All solvents were HPLC grade.

The equipment included a separation module Alliance-Bio with UV detector 2487 (Waters Inc., Milford, MA, USA) and a Pursuit C18 column (5 µm, 150 × 3.9 mm in diameter, Agilent, CA, USA). The mobile phase consisted of a 35 mM phosphate buffer pH 4.4-ACN (60:40 *v*/*v*). Compounds were separated with an isocratic flow rate of 0.8 mL/min and detected at 220 nm. Data processing was performed using Empower Pro^®^ version 2.0 software (Waters Inc.).

For the extraction procedure, for each sampling time, two circles were cut from the Guthrie card (containing 50 µL of blood each) and placed in a test tube. Then, 100 µL of 1 N sodium hydroxide solution was added and the mixture was vortexed for 30 s. Next, 3 mL of diethyl ether was added and the mixture was vortexed for 1 min and centrifuged at 800× *g* for 10 min. The tubes were frozen at −80 °C for 10 min and the organic phase was decanted and placed in a 40 °C water bath until dry, then resuspended with 190 µL of the mobile phase. Then, 10 µL of propranolol (50 µg/mL) was added as an external standard and the mixture was vortexed for 40 s and 100 µL was injected. The determination was performed in triplicate.

### 2.6. Measurement of Sedation Depth

To measure the depth of sedation in patients, a pediatric BIS™ sensor and a BIS™ Model A-2000 monitor (Covidien, Mansfield, MA, USA) were used.

Through the sensor placed on the patient’s forehead, electroencephalogram (EEG) information was obtained. The BIS system processes the EEG information and calculates a number between 0 and 100 corresponding to a direct measurement of the patient’s level of consciousness and their response to sedation. A BIS value close to 100 indicates that the patient is awake, while a BIS value of 0 indicates the absence of brain electrical activity. BIS values between 40 and 60 indicate deep sedation (Aspect Medical System, Norwood, MA, USA).

### 2.7. Pharmacokinetic Analysis

The pharmacokinetic analysis was performed using nonlinear mixed effects modeling, calculated using the Monolix Suite™ software version 2021R1 (Lixoft, Antony, France, 2021), and fitted to the bicompartmental model with first-order elimination. The variables obtained were described by the means and standard error (SE). Nonparametric rank analysis, such as the Mann–Whitney U test or Kruskal–Wallis analysis of variance, was applied according to the size and condition of the groups formed. For this purpose, the statistical package Minitab^®^ Version 19.2020.1 was used (Minitab Inc., State College, PA, USA). In all cases, *p* ≤ 0.05 was considered indicative of a statistically significant difference. Alternatively, the program SigmaPlot V.14.5 (Systat Software Inc., Palo Alto, CA, USA) was used to create graphs.

## 3. Results

Males (71/117) and patients aged 6–12 years (49/117) accounted for the majority of the study population. The group of patients group had a mean BMI between the 10th and 90th percentiles according to World Health Organization (WHO) and Centers for Disease Control (CDC) tables, indicating that the study subjects were eutrophic patients. Sixty-seven different surgical diagnoses were recorded, with the main diagnoses including cryptorchidism, appendicitis, and right microtia (Table 1).

For the data analysis, the patients were stratified into three age groups: 2–5 years and 11 months, 6–12 years and 11 months, and 13–17 years and 11 months; the pharmacokinetic parameters were determined in 351 samples.

A population PK approach with a nonlinear mixed-effect model was implemented using the Monolix Suite™ software version 2021R1 (Lixoft, Antony, France, 2021), which combines the stochastic expectation maximization algorithm and a Markov Chain Monte Carlo procedure for likelihood maximization [25,26] and is fitted to a bicompartmental model with first-order elimination. The model’s fit was visually inspected by generating visual predictive checks plots and confirmed using the log-likelihood ratio.

Each covariate was added one at a time to the model and conserved if it led to a decrease in target function (OFV); thus, no continuous covariate was included as none contributed significantly to the decrease in SFT.

The pharmacokinetic variables of clearance (CL), the volume of distribution (Vd) of the first compartment (V1), intercompartmental clearance (Q), and the Vd of the second compartment (V2) were calculated. Likewise, the maximum sedation level (Imax) was obtained as a measure of pharmacodynamic behavior. According to the results obtained, age did not influence some pharmacokinetic variables or the response to midazolam, with no significant differences found in CL, Q, or Imax. However, differences in the Vd were noted, with V1 increasing as age decreased among both sexes. V2 increased as age increased in male patients; however, in female patients, no differences were observed between age groups. With respect to the effect of sex on pharmacokinetics, differences were only found in V2 in all age ranges (Table 2).

The temporal profiles of midazolam concentrations and the degree of sedation depth by BIS values are shown in Figure 1 and Figure 2, respectively.

As shown in Figure 1, in the profiles of patients older than 6 years, the behavior of the drug was observed to be bicompartmental, with generally lower concentrations in the age group of 6–12 years; however, in the profile of patients younger than 6 years, midazolam behaved differently, showing higher levels and increases for up to 120 min and no elimination phase.

As shown in Figure 2, at the same dose of the drug, a deep and prolonged sedative effect was produced (BIS values between 40 and 60) for up to 120 min in the age group of 2–5 years, while for older subjects, BIS values increased and sedation transitioned from deep to moderate after 60 min.

Figure 3 shows the main pharmacokinetic parameters by age and sex. No differences in CL according to age and sex were found (A). Differences were found in the V1, showing an increase as age decreased among both sexes (B), and V2 showed an increase as age increased among male patients. However, among female patients, no differences were observed with respect to age. When comparing both sexes, differences were found in all age ranges (C).

## 4. Discussion

In the absence of pediatric pharmacokinetic studies to guide the safe and effective use of medications, pediatric dosing can be guided by the knowledge of anatomical and physiological factors, which help determine the distribution of drugs and the influence of these factors in this population [27]. Notably, at the usual sedation dose of 0.1 mg/kg, anesthesiologists observed that the patients were very sedated (with BIS values between 28 and 32) and, because most pediatric patients are treated in outpatient settings, the excessive sedation increased the discharge time from the postanesthetic care unit; therefore, the sedation dose was decreased to 0.05 mg/kg of midazolam—half of the initially proposed dose of 0.1 mg/kg—for greater patient safety and to avoid oversedation and adverse effects such as hypotension or cardiorespiratory depression.

Sex differences in drug metabolism are believed to have an important influence on pharmacokinetics [28].

Studies indicate that women metabolize drugs more quickly than men, which is particularly true for the substrates of the main metabolizing component of cytochrome P450, CYP3A4 [29], which may explain one of the mechanisms accounting for differences in volumes of distribution and concentrations between males and females. The distribution of the drug in children can vary compared to that of adults due to age, absorption, distribution, metabolism, and excretion [30]. These differences can result from, for example, the smaller intestine in children and altered permeability with age. In addition to gastric emptying time, intestinal transit time, bile fluid production, and blood flow to the liver and intestines may be altered in children [31]. Midazolam may be more effective in children as it has a faster onset of action and a shorter duration than other benzodiazepines [32].

The pharmacokinetics necessary to understand the action of a drug may not be closely related to the pharmacodynamic response. Although the plasma concentrations of the drug decrease rapidly, the binding of midazolam to brain receptors and other receptors can cause a disproportionate action with respect to the half-life of the drug [5]. In our study, we found that children under 6 years of age often require lower doses of midazolam (<0.05 mg/kg) to achieve the sedative effect, while children older than 6 years may require a second dose of midazolam after 120 min to remain sedated. Therefore, the dosage regimen should be adjusted in these age groups.

Some drugs, such as sedatives, have different pharmacodynamic behaviors between pediatric patients and adults, which determines particularities in their therapeutic effects [7]. The most common tool used to evaluate the depth of anesthesia is the EEG, where changes in the signals detected by the EEG correspond to anesthetic depth (spectral edge frequency, bispectrality, entropy, brain state index) [33]. The BIS showed a close relationship with the modeled concentration of propofol at the site of effect and serves as a measure of the effect of anesthetic drugs in children older than 1 year of age [34]. Several clinical studies on the effectiveness of oral midazolam in children did not identify significant effects of age [35,36]. For example, a large-scale multicenter study (397 children) found that the efficacy of midazolam did not depend on age and that a dose of 0.25 mg/kg was as effective as higher doses. These data agree with our results as we did not observe differences in the depth of sedation by the BIS in the age groups studied. The IV dose of 0.05 mg/kg midazolam administered to the patients was sufficient to obtain deep sedation (BIS values between 40 and 60) in the three age groups, with no changes in the rate of drug clearance.

Clearance of many drugs metabolized in the liver, such as midazolam [37], is increased in children (2 to 11 years) compared to adults. In our study, we did not find differences in midazolam clearance with respect to age in children aged 2–17 years.

The pharmacokinetic parameters estimated in the adult models overestimate the plasma concentrations in children. Pharmacokinetic parameters in children are influenced by size and age. The two-compartment pharmacokinetic model adequately describes midazolam’s behavior. This model is usually described using two volumes (V1, V2) and two clearance variables (CL, Q), where Q is the intercompartmental clearance and the Vd at steady state (Vss) is the sum of V1 and V2 [38], which is consistent with the pharmacokinetic analysis of our population since the bicompartmental model showed the best fit for our data.

In children, the Vd changes with age, and these changes are due to body composition (especially extracellular and total body water volumes) and plasma protein binding [39,40]. Sex differences in drug distribution also become more evident during puberty [41,42,43]. The effects of rapid growth, sexual maturation, and marked variability in pharmacokinetics should be considered during studies with adolescents [39,42,44]. The bolus dose needed to achieve a target concentration depends on the Vd and increases with decreasing age [45], which is consistent with the results of our study, where V1 values increased as age decreased in both sexes. Returning to the study of Vaughns et al. [40], this phenomenon may have been observed because children from 1 year of age have a higher percentage of fat and a lower percentage of protein (22.4% fat and 13.4% protein) than older children (13% fat and 18.1% protein).

A decrease in binding due to a lower percentage of proteins can increase the already high Vd of midazolam by allowing more rapid and extensive redistribution. Changes in the size of biological fluid compartments and alterations in regional blood flow and membrane permeability can also alter the rate and extent of redistribution. Body fat, which will bind to a lipid-soluble drug, increases with age and will increase the Vd [5]. The latter phenomenon is consistent with the V2 values that we obtained in male patients, which increased with age. Studies in the literature report that fat-soluble drugs, such as midazolam, are more widely distributed in women than in men [1] as women are smaller and thus have more fat, less muscle, and less body water [4]. We did not observe this in our study, perhaps because, as mentioned previously, males predominated in all age groups, and we did not have the same number of female patients to observe differences that may exist between the two sexes.

More studies are needed to address the effects of age and sex on the pharmacokinetics and pharmacodynamics of sedative agents such as midazolam due to their relevance to pharmacotherapy in the area of pediatric anesthesiology.

## 5. Conclusions

In our study, for patients younger than 6 years, lower IV midazolam doses of 0.05 mg/kg are likely required to achieve deep sedation, while for patients older than 6 years, a second dose of the drug may be required in surgeries lasting more than 120 min to maintain deep sedation. Therefore, the dosage regimen should be adjusted for these age groups, especially for male patients, who may require higher IV midazolam doses of 0.05 mg/kg when undergoing minor surgeries. Supervision is suggested to ensure the safe use of drugs in pediatric anesthesia.

## Figures and Tables

**Figure 1 pharmaceutics-15-00440-f001:**
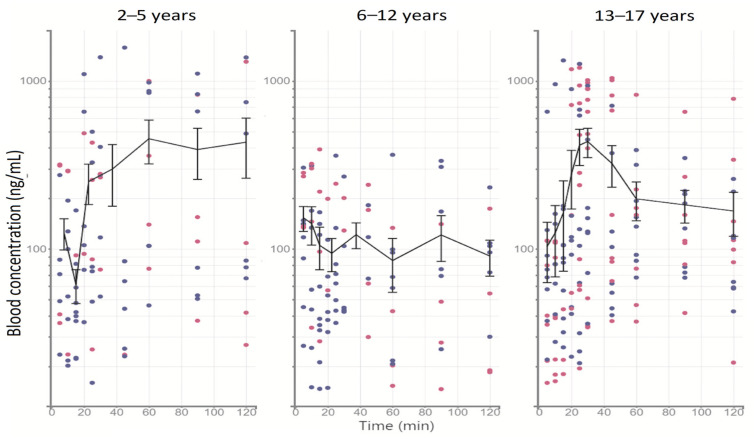
Pharmacokinetic profiles distributed by age range. The line describes the arithmetic mean ± SE. Blue points indicate male patients and pink points indicate female patients.

**Figure 2 pharmaceutics-15-00440-f002:**
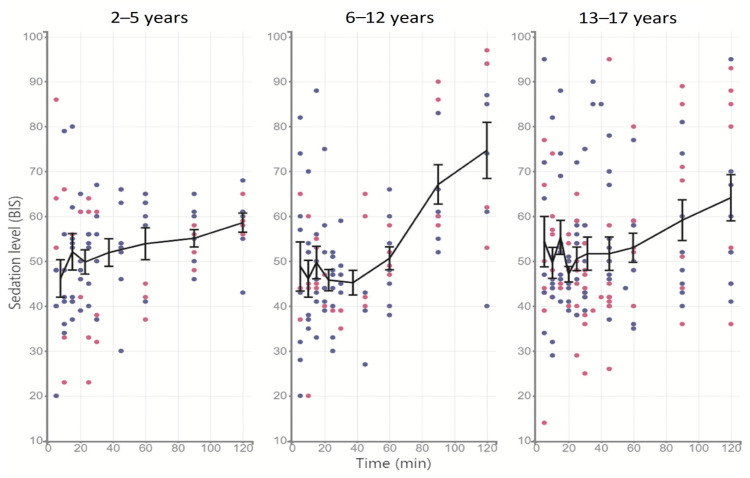
Temporal courses of sedation levels (BIS) distributed by age range; the line describes the arithmetic mean ± SE. Blue points indicate male patients and pink points indicate female patients.

**Figure 3 pharmaceutics-15-00440-f003:**
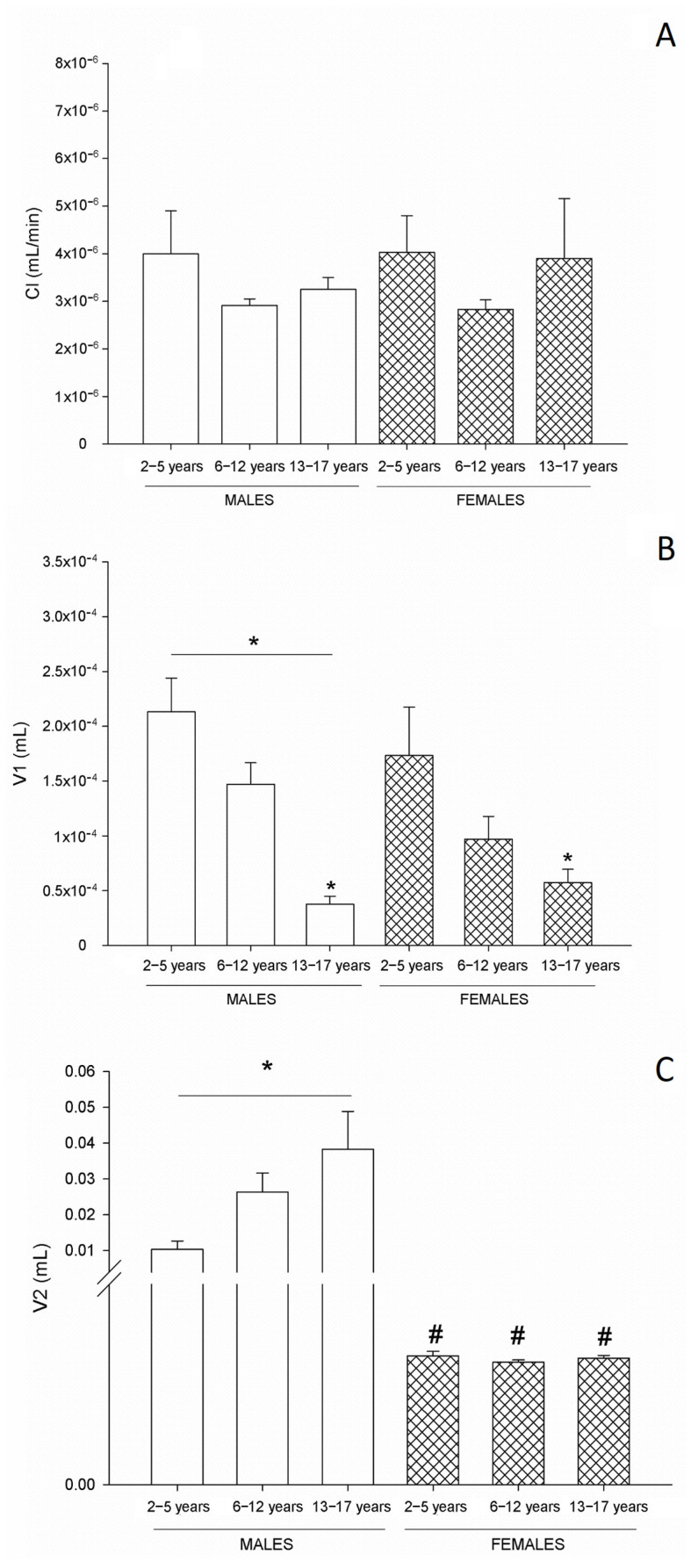
Main pharmacokinetic parameters by age and sex for patients undergoing minor surgeries. The symbol * corresponds to *p* < 0.05 values according to age range, while the symbol # refers to *p* < 0.05 values according to sex. Kruskal–Wallis analysis of variance followed by Dunn’s multiple comparisons test.

**Table 1 pharmaceutics-15-00440-t001:** Demographic characteristics of the patients undergoing minor surgeries.

Characteristics	Total (n = 117)
Number of patients (n)	
2–5 years	31
6–12 years	49
13–17 years	37
Sex (male/female)	
2–5 years	21/10
6–12 years	28/21
13–17 years	22/15
Age (years) *	
2–5 years	4 (3–5)
6–12 years	10 (8–12)
13–17 years	15 (13–16.5)
BMI (kg/m^2^) *	
2–5 years	15.5 (14–17.2)
6–12 years	19.0 (16.1–21.8)
13–17 years	22.3 (19.5–24.5)
Main diagnoses n (%)	
1. Cryptorchidism	12 (10%)
2. Appendicitis	10 (9%)
3. Right microtia	7 (6%)
4. Septal deviation	4 (3%)
5. Hypospadias	4 (3%)
6. Cleft lip	4 (3%)
7. Other	76 (66%)

* Values expressed as the median (interquartile range (Q_25_–Q_75_)). BMI expressed in values between the 10th and 90th percentiles = Eutrophic (WHO, CDC).

**Table 2 pharmaceutics-15-00440-t002:** Pharmacokinetic variables (CL, V_1_, Q, and V_2_) and pharmacodynamics (I_max_) by age range and sex.

Age	Sex	CL (L/min)	V_1_ (L)	Q (L/min)	V_2_ (L)	I_max_ (BIS)
2–5 years	Male	4.00 × 10^−6^ ± 4.15 × 10^−6^	2.13 × 10^−4^ ± 1.41 × 10^−4^	1.84 × 10^−3^ ± 4.00 × 10^−4^	1.03 × 10^−2^ ± 1.09 × 10^−2^	52.17 ± 7.76
	n = 21	(6.1 × 10^−7^–2.0 × 10^−5^)	(3.20 × 10^−5^–5.60 × 10^−4^)	(3.90 × 10^−4^–2.50 × 10^−3^)	(4.20 × 10^−4^–4.50 × 10^−2^)	(38.18–68.46)
	Female	4.03 × 10^−6^ ± 2.44 × 10^−6^	1.73 × 10^−4^ ± 1.40 × 10^−4^	1.93 × 10^−3^ ± 2.11 × 10^−4^	1.93 × 10^−3^ ± 2.11 × 10^−4^ #	51.80 ± 10.42
	n = 10	(4.8 × 10^−7^–8.1 × 10^−6^)	(3.70 × 10^−5^–5.00 × 10^−4^)	(1.60 × 10^−3^–2.30 × 10^−3^)	(1.60 × 10^−3^–2.30 × 10^−3^)	(32.89–69.08)
6–12 years	Male	2.91 × 10^−6^ ± 7.10 × 10^−7^	1.47 × 10^−4^ ± 1.04 × 10^−4^	1.81 × 10^−3^ ± 2.24 × 10^−4^	2.64 × 10^−2^ ± 2.79 × 10^−2^	52.23 ± 8.96
	n = 28	(2.0 × 10^−6^–4.5 × 10^−6^)	(1.5 × 10^−5^–4.90 × 10^−4^)	(9.50 × 10^−4^–2.10 × 10^−3^)	(3.40 × 10^−3^–1.40 × 10^−1^)	(39.39–73.63)
	Female	2.83 × 10^−6^ ± 9.43 × 10^−7^	9.70 × 10^−5^ ± 9.50 × 10^−5^	1.83 × 10^−3^ ± 1.62 × 10^−4^	1.83 × 10^−3^ ± 1.62 × 10^−4^ #	54.25 ± 11.81
	n = 21	(1.40 × 10^−6^–5.00 × 10^−6^)	(8.90 × 10^−5^–3.8 × 10^−4^)	(1.50 × 10^−3^–2.10 × 10^−3^)	(1.50 × 10^−3^–2.10 × 10^−3^)	(41.61–76.8)
13–17 years	Male	3.25 × 10^−5^ ± 1.17 × 10^−6^	3.75 × 10^−5^ ± 3.45 × 10^−5^ *	1.83 × 10^−3^ ± 1.88 × 10^−4^	3.83 × 10^−2^ ± 4.96 × 10^−2^ *	56.30 ± 11.01
	n = 22	(2.2 × 10^−6^–7.5 × 10^−6^)	(4.80 × 10^−6^–1.2 × 10^−4^)	(1.50 × 10^−3^–2.10 × 10^−3^)	(3.10 × 10^−3^–2.30 × 10^−1^)	(41.91–74.69)
	Female	3.90 × 10^−6^ ± 4.88 × 10^−6^	5.72 × 10^−5^ ± 4.69 × 10^−5^ *	1.89 × 10^−3^ ± 1.67 × 10^−4^	1.89 × 10^−3^ ± 1.67 × 10^−4^ #	51.57 ± 11.17
	n = 15	(6.10 × 10^−7^–2.00 × 10^−5^)	(7.70 × 10^−6^–1.90 × 10^−4^)	(1.50 × 10^−3^–2.20 × 10^−3^)	(1.50 × 10^−3^–2.20 × 10^−3^)	(34.81–73.84)

Data are presented as the mean ± SE; range in parentheses. * *p* < 0.05 according to age range, # *p* < 0.05 according to sex; Kruskal–Wallis analysis of variance followed by Dunn’s multiple comparisons test.

## Data Availability

The data presented in this study are available upon request from the corresponding author. The data are not publicly available due to ethical restrictions as the information could compromise the confidentiality and privacy of research participants.

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
