# Peer review of "Influence of Age and Sex on the Pharmacokinetics of Midazolam and the Depth of Sedation in Pediatric Patients Undergoing Minor Surgeries"

_pharmaceutics, 2023, doi:10.3390/pharmaceutics15020440_

Round 1

Reviewer 1 Report

The authors conducted detailed research on the pharmacokinetics of midazolam with different groups of age and sex in children. The concentrations were measured by HPLC, and the sedation depth was measured by BIS sensor and monitor. The results suggests that a second dosage is required for children who are above 6 years old, especially for male patients. However, it would be wise to apply this conclusion based on the individuals since the metabolisms are various among individuals. There are some comments as below:

1.      Midazolam can be metabolized into several metabolites, for instance, its main metabolite 1-hydroxymidazolam. Do the authors see from HPLC and what is the half life?

2.      What are the time intervals for blood withdraw?

3.      Whether the HPLC analysis provided a standard for midazolam?

4.      In Table 2, it looks like CL slightly decreases by the increasing of the age (from 2-5 group to 6-12 group), and then increases again (13-17), not exactly as the authors indicated there is no difference.

5.      Please describe the instrument and method which were used for HPLC since this is the major method for this study.

Reviewer 2 Report

The sedative effect and the pharmacokinetic properties of midazolam are analyzed in this study in various pediatric age groups and in each gender. Based on their measurements and pharmacokinetic modelling, the authors conclude that <0.05 mg/kg doses are required in pediatric patients younger than 6 years, while another dose is required in older patients if deep sedation should continue for more than 120 min. Midazolam was assayed in dried blood spots prepared from heparinized blood collected in a phlebotomy process, using high-performance liquid chromatography.

The manuscript is well written, and it is certainly important to support clinical decisions regarding sedation with a clinical pharmacologist’s perspective and with evidence. Nevertheless, I recommend the revision of the manuscript.

 Major remarks:

1. The most substantial piece of information that is missing is the clinical importance of this study. What is the consequence of under - or overdosing children of various ages with midazolam? Are there any special patient populations who may be affected especially severly by being dosed erroneously? Please provide a detailed yet specific explanation of this in the Introduction.

2.   In the Introduction, please explain how midazolam is used in comparison to other benzodiazepines regarding the sedation of pediatric patients, along with the risks and benefits.

3. Please describe the advantage of preparing dried blood spots. This introduces a source of unreliability to the measurements in comparison to assaying plasma, especially when blood is collected on filter paper.

4. Provide data on the analytical performance of the HPLC method when used with dried blood spot samples. Consider the work of Capiau et al (https://doi.org/7/FTD.0000000000000643) to this end.

5. Provide explanation on the rationale of using the BIS score with appropriate reference.

6. You are requested to explain why a bicompartmental model was used by comparing its performance to that of a unicompartmental model, or by displaying concentrations on a semilogarithmic plot. The use of a bicompartmental model is not supported by the fact that only three samples were collected from each patient. Provide evidence of avoiding overfitting. In addition, please explain how you have tested candidate covariates for inclusion and why you have not included any.

7. Please explain your observational findings in more detail.

a.      Include the statistical description of the clinical information collected regarding the subjects of the study (lines 78-80).

b.      Please provide evaluation of the adverse effects observed and make a comparison between various age groups and between males and females.

c.      In Figure 2, individual data are presented along with arithmetic means and SD’s. Is there evidence that these data are Gaussian? If there is not any, please consider displaying boxplots (indicating medians and interquartile ranges) instead.

d.      The longitudinality of the data presented is not clear. Please explain how the longitudinal evaluation was performed and display the longitudinal data series.

8. Only 7 of the 33 references are from 2017 or later, with several dating back to the 1990’s or even 1980’s. Please revise the list of cited articles to include the majority from the past five years.

Minor remarks:

9. In Figure 3, what do * and # correspond to?

10. In line 308, do you mean a lower dose of, or than 0.05 mg/kg?

11. Starting with line 304, the authors claim that „More studies are needed to address the effects of age and sex on the pharmacokinetics and pharmacodynamics of sedative agents, such as midazolam, due to their relevance to pharmacotherapy in the area of paediatric anaesthesiology”. Please explain the limitations of your study which prompt this conclusion, i.e. why your results cannot be used directly by clinicians to guide midazolamolam administration.
